

# Biases in genome reconstruction from metagenomic data

William C. Nelson[1], Benjamin J. Tully[2,3] and Jennifer M. Mobberley[4]

[1] Biological Sciences Division, Pacific Northwest National Laboratory, Richland, WA, USA
[2] Department of Biological Sciences, Marine Environmental Biology Section, University of Southern California, Los Angeles, CA, USA
[3] Center for Dark Energy Biosphere Investigations, University of Southern California, Los Angeles, CA, USA
[4] Chemical and Biological Signature Science Group, Pacific Northwest National Laboratory, Richland, WA, USA

## ABSTRACT

**Background:** Advances in sequencing, assembly, and assortment of contigs into species-specific bins has enabled the reconstruction of genomes from metagenomic data (MAGs). Though a powerful technique, it is difficult to determine whether assembly and binning techniques are accurate when applied to environmental metagenomes due to a lack of complete reference genome sequences against which to check the resulting MAGs.

**Methods:** We compared MAGs derived from an enrichment culture containing ~20 organisms to complete genome sequences of 10 organisms isolated from the enrichment culture. Factors commonly considered in binning software—nucleotide composition and sequence repetitiveness—were calculated for both the correctly binned and not-binned regions. This direct comparison revealed biases in sequence characteristics and gene content in the not-binned regions. Additionally, the composition of three public data sets representing MAGs reconstructed from the *Tara* Oceans metagenomic data was compared to a set of representative genomes available through NCBI RefSeq to verify that the biases identified were observable in more complex data sets and using three contemporary binning software packages.

**Results:** Repeat sequences were frequently not binned in the genome reconstruction processes, as were sequence regions with variant nucleotide composition. Genes encoded on the not-binned regions were strongly biased towards ribosomal RNAs, transfer RNAs, mobile element functions and genes of unknown function. Our results support genome reconstruction as a robust process and suggest that reconstructions determined to be >90% complete are likely to effectively represent organismal function; however, population-level genotypic heterogeneity in natural populations, such as uneven distribution of plasmids, can lead to incorrect inferences.

Corresponding author
William C. Nelson,
william.nelson@pnnl.gov

## INTRODUCTION

High-throughput sequencing has revolutionized microbiology by circumventing "the great plate count anomaly" (*Staley & Konopka, 1985*) and allowing direct investigation of

natural communities in a culture-independent manner (*Venter et al., 2004*; *DeLong et al., 2006*; *Costello et al., 2009*; *Caporaso et al., 2012*; *Zhou et al., 2015*; *Rinke et al., 2013*; *White et al., 2016*). One goal of metagenomics has always been to obtain organism-specific, complete, genomic information from the complex mixture of sequence data generated from environmental samples. Having a complete genome sequence provides a platform for understanding the range of metabolic roles an organism can play within a community and the interactions it has with other organisms (*Iverson et al., 2012*; *Sharon et al., 2013*; *Delmont et al., 2015*), and it can provide specific context for interpretation of transcriptomics and proteomics (*Lesniewski et al., 2012*; *Ram et al., 2005*). Metagenome-assembled genomes (MAGs) are produced by segregating assembled contigs/scaffolds into organism-specific "bins". This process of genome reconstruction has benefited from continuing advances in sequencing technologies, sequence assembly algorithms, and segregation methods (*Sangwan, Xia & Gilbert, 2016*). Early success assembling genomes from a simple community (*Tyson et al., 2004*) has led to more recent studies reconstructing many organisms from complex environments (*Brown et al., 2015*; *Anantharaman, Breier & Dick, 2016*; *Baker et al., 2015*; *Li et al., 2015*; *Nobu et al., 2015*; *Tully, Graham & Heidelberg, 2018*; *Parks et al., 2018*; *Pasolli et al., 2019*; *Almeida et al., 2019*; *Mobberley et al., 2017*; *Stewart et al., 2019*; *Pedron et al., 2019*; *Wong et al., 2018*; *Daly et al., 2016*; *Danczak et al., 2017*). The accuracy of these techniques in the context of a complex environmental community is difficult to gauge; however, because most available complete microbial genome sequences that could serve as references are from cultured isolates, and these isolates are rarely present in environmental metagenomes. Techniques that have been developed to evaluate the accuracy of the binning process rely on conserved genes and consistency of nucleotide composition (*Eren et al., 2015*; *Parks et al., 2015*; *Waterhouse et al., 2018*; *Chen et al., 2019*; *Hugoson, Lam & Guy, 2019*). These techniques, however, cannot make accurate determinations of how much sequence is missing or the functional potential of missing content. Genome reconstruction techniques have been tested using synthetic communities of cultured organisms (*Hardwick et al., 2018*) and simulated metagenomic datasets. Over time, increasingly sophisticated methods have been developed to simulate metagenomic read data sets, from the earlier Grinder (*Angly et al., 2012*), MetaSim (*Richter et al., 2008*), GemSIM (*McElroy, Luciani & Thomas, 2012*), BEAR (*Johnson et al., 2014*), and NeSSM (*Jia et al., 2013*), to the more recent CAMISIM (*Fritz et al., 2019*), which was developed as part of the community effort to address standards in metagenome analysis software development (*Sczyrba et al., 2017*). Generally these simulators concern themselves with modeling community structure and sequencing attributes, such as read length and error rates, but are limited to presenting data generated from a reference genomic database, thus cannot model the genetic diversity found in most environments, although CAMISIM addresses this issue by implementing the genome evolution simulator sgEvolver (*Darling, 2004*). Because genetic variability within natural populations is, as yet, ill-defined (*Rocha, 2018*), it is unlikely that such test data can accurately replicate the type and amount of variability found in natural communities, and the complications this variability causes.

Unicyanobacterial consortia (UCC) were developed as model systems to investigate the mechanisms of metabolic interaction between cyanobacteria and heterotrophs. These systems provide an opportunity to compare MAGs against a matching reference genome set and learn about potential gaps and pitfalls of current reconstruction processes. Two consortia, each containing a single unique cyanobacterial species and sharing an additional 18 heterotrophic species, were derived from a natural mat community (*Cole et al., 2014*). The communities have been sequenced, and genome reconstruction has been performed (*Nelson et al., 2015*), yielding near-complete genome sequences revealing the presence and maintenance of microdiversity, such as might be found within an intact environmental sample. Thus, this system more accurately reflects in situ community diversity compared to synthetic communities constructed from isolated organisms. In parallel, isolates of 10 of the member species have also been sequenced (*Nelson et al., 2015*; *Romine et al., 2017*). This paired genomic and metagenomic data set allows direct comparison of MAGs from diverse organisms against "ground truth" genomic data. Previously, we have shown that common aspects of the genome reconstruction process (assembly from a complex sequence space and segregation of contigs based on read depth profiles and sequence composition) to be both specific and sensitive (*Nelson et al., 2015*).

We have investigated the nature of genomic regions that under current standard genome reconstruction techniques are not recovered (herein referred to as *not-binned regions*, or *NRs*) to evaluate how these regions differ from recovered regions (*correctly binned regions*, or *CRs*), and to what extent the missing genomic information might impact conclusions drawn from analysis of MAGs. Two common elements of current sequence segregation protocols are analysis of sequence composition and comparison of coverage profiles between samples, so we compared the nucleotide content of NRs vs CRs, examining both %G+C and tetranucleotide content, and the redundancy of sequence information both within the individual genome (i.e., repetitiveness within the genome) and across the entire metagenomic data set (i.e., sequence shared between populations). To determine the impact on downstream functional analyses, the gene content was examined for biases in the cellular roles of genes found within NRs and CRs. To verify that the biases observed extended to more complex metagenomic datasets and across binning algorithms, the *Tara* Oceans metagenome, which has been binned by different groups using MetaBAT (*Parks et al., 2018*; *Kang et al., 2015*), Anvi'o (*Eren et al., 2015*; *Delmont et al., 2018*), and BinSanity (*Tully, Graham & Heidelberg, 2018*; *Graham, Heidelberg & Tully, 2017*), was subjected to similar sequence and repeat compositional analysis.

## MATERIALS AND METHODS

### Data and code availability

The UCC MAG and genome data analyzed are available in the GenBank repository as listed in Table 1. The metagenomic data used to construct the UCC MAGs is available from the NCBI SRA (accessions SRX1063989 and SRX1065184). MAGs reconstructed from the *Tara* Oceans metagenomic data (*Tully, Graham & Heidelberg, 2018*; *Parks et al., 2018*) are available in the GenBank repository. MAGs from *Delmont et al. (2018)* are
**Table 1 Reconstructed genome coverage and completeness.**

| Genome | Molecule identifier | Genome NCBI accessions | MAG NCBI accessions | MG Cov[a] | %CR[b] | NR[c] | Mean NR length (bp) | NR length range |
|--------|--------------------|-----------------------|---------------------|-----------|--------|-------|--------------------|-----------------|
| HL-46 | EI34DRAFT_7210 | JQMU01000001.1 | GCA_001314525.1 | 3.9× | 40 | 284 | 4,742 | 1,007..42,318 |
|  | EI34DRAFT_6181[d] | JQMU01000002.1 |  | 3.9× | 25 | 7 | 18,136 | 1,108..49,149 |
| HL-48 | CY41DRAFT | KK366039.1 | GCA_001314875.1 | 69× | 95 | 29 | 1,892 | 330..53,737 |
| HL-49 | K302DRAFT | JAFX01000001.1 | GCA_001314815.1 | 9.7× | 91 | 89 | 3,234 | 209..25,366 |
| HL-53 | Ga0003345 | LN899469.1 | GCA_001314555.1 | 113× | 98 | 15 | 1,564 | 952..6,133 |
| HL-55 | K417DRAFT | JYNR01000001.1 | GCA_001314845.1 | 11× | 95 | 34 | 3,574 | 417..45,387 |
| HL-58 | CD01DRAFT | JMLY01000001.1 | GCA_001314605.1 | 128× | 99 | 13 | 1,124 | 959..12,996 |
| HL-91 | Ga0058931_14 | FBYC01000004.1 | GCA_001314645.1 | 226× | 97 | 20 | 3,129 | 135..11,341 |
|  | Ga0058931_11[d] | FBYC01000001.1 |  | 227× | 97 | 6 | 2,188 | 914..4,391 |
|  | Ga0058931_13[d] | FBYC01000003.1 |  | 158× | 0 | 1 | 113,349 | 113,349 |
|  | Ga0058931_12[d] | FBYC01000002.1 |  | 160× | 0 | 1 | 97,917 | 97,917 |
| HL-93 | Ga0071314 | LT593974.1 | GCA_001314745.1 | 11× | 85 | 98 | 3,605 | 232..78,515 |
| HL-109 | Ga0071312_11 | FMBM01000001.1 | GCA_001314785.1 | 612× | 87 | 20 | 1,835 | 204..63,971 |
|  | Ga0071312_12 | FMBM01000002.1 |  | 669× | 92 | 28 | 1,285 | 506..52,589 |
|  | Ga0071312_13[d] | FMBM01000003.1 |  | 615× | 95 | 3 | 6,053 | 1,908..10,088 |
| HL-111 | Ga0071316 | LT629743.1 | GCA_001314765.1 | 18× | 95 | 39 | 1,589 | 501..20,407 |

**Notes:**
[a] Metagenomic read coverage.
[b] Percentage of the genome represented in the MAG.
[c] Number of not-binned regions.
[d] Predicted to be an extrachromosomal element.

available through figshare (DOI 10.6084/m9.figshare.4902923). A list of MAGs and corresponding identifiers are available in Table S1. Complete bacterial and archaeal genomes were collected from NCBI RefSeq (O'Leary et al., 2016) (accessed Aug 2019) based on assignment as either "reference genome" or "representative genome" for the data column "refseq_category" and "Complete Genome" in the "assembly_level" column. A list of genomes used in the analysis are available in Table S2. All analysis scripts are available at http://github.com/wichne/biases_in_genome_reconstruction.

### Identification of CR and NR regions

The UCC scaffolds comprising each MAG were searched against their cognate complete genome sequence using nucmer using the maxmatch option (Kurtz et al., 2004). Regions of the genomes that aligned end-to-end to MAG scaffolds at ≥99% identity were cataloged as CR regions. All other genome regions were considered NR regions.

### Compositional analysis

For the UCC MAGs and genomes, %G+C calculation and tetranucleotide frequency (TNF) chi-square test were performed using custom Perl scripts (available at http://github.com/wichne/biases_in_genome_reconstruction). Compositional analysis was restricted to CR or NR regions longer than 1,000 bp to ensure sufficient sequence for meaningful results. For TNF, the chi-squared statistic was calculated for each region using the TNF for the whole genome as the expected values, and the mean and standard deviation for the

CR and NR pools calculated. For %G+C analysis, the mean %G+C for the CR and NR regions was calculated, and the absolute difference was calculated between each region and the genome average, and average differences determined for CR and NR pools. To estimate $p$-values for the %G+C and TNF analyses, one thousand random coordinate sets yielding the same number and length of fragments as in each genome's CR or NR set were generated from the genome sequence and evaluated.

For comparison of the UCC data set to the *Tara* Oceans MAGs and RefSeq genome data sets, sequence composition variance (i.e., deviation from the mean) was calculated for the %G+C and tetranucleotide frequency using a custom Python script. The %G+C was calculated for 2 kb segments (sliding window of 500 bp) for each MAG or genome. A genome-wide variance value was calculated for each MAG or genome based on the segments and plotted as a box plot per source data set. TNF was calculated for 10 kb segments (sliding window 5 kb) for each MAG or genome. Using the calculation described in *Teeling et al. (2004)*, each segment had a $Z$-score calculated for each tetranucleotide based on the observed-vs-expected frequency of the tetranucleotide in the 10 kb segment. A Pearson correlation was then calculated in a pairwise fashion for all segments. Variance of the Pearson correlation values within a MAG or genome was calculated and plotted as a box plot per source data set.

## Repetitiveness analysis

To calculate intragenome sequence repetitiveness, we determined the fraction of each genome that was comprised of repeat sequence. Each genome sequence was searched against itself using nucmer v3.0 (*Kurtz et al., 2004*) with the maxmatch option, and the lengths of regions that aligned to another part of the genome/MAG with ≥97% identity were summed and divided by the length of the genome/MAG.

To determine the repetitiveness of sequences across the entire metagenomic data set, metagenome reads were searched against genome sequences using Bowtie2 (*Langmead & Salzberg, 2012*). Per-base coverage was calculated using the samtools (*Li et al., 2009*) depth command, and average coverage values for the genomes, NRs and CRs were determined. One thousand sets of random coordinate regions of the same number and lengths as in each set were analyzed to estimate $p$-values. Results are reported as average coverage depth of NRs and CRs and the average difference from the genome depth-of-coverage.

## Gene function analysis

Unicyanobacterial consortial cultures complete genome sequences were annotated by the IMG pipeline (*Huntemann et al., 2015*), which included COG assignment based on the December 2014 release of the 2003–2014 COGs (*Galperin et al., 2015*). COGs assigned to more than one functional category were counted for each assigned category. Genes not assigned to a COG category were classified as "unassigned". Ribosomal RNA (rRNA) gene features were identified by the IMG pipeline (*Markowitz et al., 2014*); transfer RNAs (tRNA) were identified with tRNAscan-SE (*Lowe & Eddy, 1997*); other non-coding RNAs (ncRNA) were identified using the Rfam database v11.0 (*Burge et al., 2013*) and

infeRNAl v1.1 software (*Nawrocki & Eddy, 2013*). For each gene set, the category counts were normalized to the total feature counts. Principle component analysis was performed and biplot of gene categories was generated using R package bpca v.1.2-2 (http://cran.r-project.org/web/packages/bpca/).

## Statistical analysis

Statistical tests were performed using modules within the Python package SciPy (*Virtanen et al., 2019*). The normality of the calculated variance distributions for each set of genomes was determined using the Shapiro–Wilk test (*Shapiro & Wilk, 1965*). Genome sets with a normal distribution were compared to each other with the *T*-test for two independent variables (*Welch, 1947*). Genome sets without a normal distribution were compared to each other with the Mann–Whitney U test (*Mann & Whitney, 1947*). *p*-values were adjusted for multiple comparisons with the Benjamini-Hochberg procedure (*Benjamini & Hochberg, 1995*) correction with a false discovery rate of 25% (Table S3).

## RESULTS AND DISCUSSION

The power of metagenomics is that it allows exploration of diverse communities from which we cannot culture the component populations either because the proper growth conditions are unknown or difficult to replicate in a laboratory environment, or simply because there are too many organisms present to have the resources or time to pursue the effort. Because of this, there are very few examples of sequenced organisms isolated from the same sample from which metagenomic sequencing and binning has been done to generate MAGs. As such, a "gold standard" for evaluation of MAG content has been difficult to come by. We have taken advantage of two enrichment cultures from which MAGs and isolate genomes have been derived to generate just such a "gold standard" comparison framework. We have previously generated two unicyanobacterial consortial cultures (UCC) – enrichment cultures each containing a distinct cyanobacterial population and different, yet overlapping, communities of associated heterotrophs, each numbering <20 species—and performed metagenomic sequencing, assembly and binning (*Nelson et al., 2015*; *Romine et al., 2017*). Illumina 150 bp paired-end reads were generated from each community, and IDBA_ud was used to assemble the read sets separately and in co-assembly. The abundances of the organisms differed between the two communities, allowing us to bin the sequences by comparing sequence coverage values of contigs between the two UCCs in a predominantly manual process (inspired by the work of *Dick et al. (2009)*). The resulting MAGs were manually curated to eliminate contaminating contigs and identify mis-binned contigs, correctly placing them when possible. In parallel, ten organisms were isolated from the UCCs and completely sequenced. Comparison of the MAGs to the isolate genomes showed recovery of >90% of sequence for genomes with at least 10× coverage, with one exception, *Halomonas* sp. HL-93, which had 85% recovery from 11× coverage (Table 1). Co-linear sequence alignments indicated there were no assembly errors in the binned contigs (*Nelson et al., 2015*, and data not shown). Based on the isolate-MAG comparisons, NRs were identified. *Porphyrobacter* HL-46 had the lowest metagenome coverage (3.6×). Its MAG comprised hundreds of short contigs

and was determined to be ~40% complete. Thus, the NRs for HL-46 are assumed to be primarily caused by the random sampling of the shotgun sequencing methodology and not by any inherent content biases, allowing the HL-46 analyses to serve as a control.

To determine if NRs were not binned due to lack of assembly, we mapped the contigs from the assembly to the CR and NR regions of the genomes and looked at the contig coverage of the regions. As expected, the CRs showed an average contig coverage of $1.04 \pm 0.14$, and most regions had only a single contig map to them (Fig. S1). Many of the cases of multiple contigs mapping to a CR were due to short (<200 bp) contigs of repeat sequence which might be an artifact of the assembler (IDBA_ud). NRs show a strong positive correlation between region length and number of contigs mapping, with an average coverage of $0.94 \pm 0.71$ (Fig. S2). This suggests poorer assembly of the NRs and higher repeat content, but also indicates that most NR sequence is present in the contig set, and thus the binning process is the main determinant of NRs.

## Nucleotide composition of NRs frequently differs from the genome average

Bacteria and Archaea have evolved to have a fairly consistent %G+C across their genome (*Karlin, Campbell & Mrazek, 1998*), so much so that it has been proposed as a metric of classification at higher taxonomic levels (*Wayne et al., 1987*). It is not uncommon, however, to observe regions within a genome that differ significantly from the genome average (*Bohlin et al., 2010*). This variation can be the result of selective pressure for structural properties in non-coding genes, for instance ribosomal RNAs and other functional RNAs have been shown to vary in nucleotide composition in correlation with optimal growth temperature (*Galtier & Lobry, 1997*). In other cases, divergent %G+C indicates a region which has been acquired recently (in evolutionary time) from a non-related source (i.e., horizontal gene transfer) (*Wixon, 2001*). To investigate whether variant G+C confounds genome reconstruction, we compared the %G+C of NRs to that of CRs and the complete genome.

The genomes in this study had a range of %G+C values, from 42% (*A. marincola* HL-49) to 68% (*Erythrobacteraceae* bacterium HL-111), with most skewing toward the higher values (Table 2). We determined the %G+C for each CR and NR ≥200 bp in length and compared them to the %G+C for the complete genome. For genomes with more than one genomic element, each molecule was considered separately since extrachromosomal elements may have distinct nucleotide composition. For seven of the genomes, the %G+C for the NRs differed significantly ($p \leq 0.005$) from the genome average, while the CRs generally reflected the genome average (Table 2). The %G+C averages for NRs from HL-48 and HL-111 were significantly lower (45.76% and 64.26%, respectively) than the genomes' averages (58.98% and 68.12% respectively). Other genomes (HL-53, HL-55, HL-109) had some NRs with %G+C higher than the genome average and some NRs with lower values (Fig. 1), despite having different average %G+C values (47.5%, 56.0% and 64.1% respectively). Extrachromosomal elements analyzed did not display a significant difference in the %G+C of their NRs from the molecule average. As expected, the values for the NRs and CRs of HL-46 showed no significant

**Table 2  %G+C analysis.**

| | Molecule | Genome | CRs | | | NRs | | |
|---|---|---|---|---|---|---|---|---|
| | | Mean | Mean | Distance | *p*-value | Mean | Distance | *p*-value |
| HL-46 | EI34DRAFT_7210 | 64.42 | 63.96 ± 1.94 | 1.55 ± 1.25 | 0.997 | 65.12 ± 2.13 | 1.61 ± 1.56 | 0..263 |
| HL-46 | EI34DRAFT_6181 | 59.94 | 60.78 ± 2.27 | 1.97 ± 1.41 | 0.856 | 60.97 ± 1.78 | 1.92 ± 0.75 | 0.605 |
| HL-48 | CY41DRAFT | 58.98 | 59.00 ± 1.52 | 1.01 ± 1.13 | 0.996 | 45.76 ± 19.69 | 13.22 ± 19.69 | **<0.001**[a] |
| HL-49 | K302DRAFT | 42.22 | 42.24 ± 1.71 | 1.15 ± 1.27 | 0.434 | 42.73 ± 3.37 | 2.44 ± 2.38 | **0.001** |
| HL-53 | Ga0003345 | 47.50 | 46.95 ± 1.61 | 0.96 ± 1.40 | 0.031 | 48.83 ± 3.55 | 3.70 ± 0.82 | **<0.001** |
| HL-55 | K417DRAFT | 56.26 | 55.87 ± 1.97 | 1.42 ± 1.41 | 0.025 | 55.44 ± 3.30 | 3.00 ± 1.59 | **0.001** |
| HL-58 | CD01DRAFT | 57.56 | 56.83 ± 2.61 | 1.69 ± 2.12 | 0.047 | 56.11 ± 3.69 | 3.93 ± 0.51 | 0.016 |
| HL-91 | Ga0058931_11 | 61.75 | 62.05 ± 0.25 | 0.31 ± 0.23 | 0.954 | 60.39 ± 3.17 | 2.79 ± 2.02 | 0.053 |
| HL-91 | Ga0058931_12 | 60.37 | nd[b] | nd | nd | nd | nd | nd |
| HL-91 | Ga0058931_13 | 61.77 | nd | nd | nd | nd | nd | nd |
| HL-91 | Ga0058931_14 | 61.84 | 60.99 ± 1.90 | 1.33 ± 1.60 | 0.030 | 59.11 ± 2.96 | 3.52 ± 1.96 | **0.005** |
| HL-93 | Ga0071314_11 | 55.88 | 56.75 ± 2.20 | 1.75 ± 1.59 | 1.000 | 56.08 ± 4.42 | 3.6 ± 2.57 | **<0.001** |
| HL-109 | Ga0071312_11 | 64.09 | 64.55 ± 1.46 | 1.12 ± 1.05 | 0.715 | 60.96 ± 3.02 | 3.28 ± 2.85 | 0.073 |
| HL-109 | Ga0071312_12 | 64.07 | 63.89 ± 1.41 | 0.92 ± 1.09 | 0.169 | 63.11 ± 2.21 | 1.94 ± 1.43 | 0.593 |
| HL-109 | Ga0071312_13 | 65.34 | 65.47 ± 0.07 | 0.13 ± 0.07 | 0.778 | 61.68 ± 2.24 | 3.66 ± 2.24 | 0.009 |
| HL-111 | Ga0071316_11 | 68.12 | 68.20 ± 1.44 | 0.99 ± 1.05 | 0.465 | 64.26 ± 1.39 | 3.86 ± 1.39 | **<0.001** |

**Notes:**
[a] Bold type indicates significant results ($p \leq 0.005$).
[b] Not determined because the entire molecule was missing from the reconstructed genome.

difference from the genome average (Table 2), however, HL-46's CRs and NRs did not display identical %G+C profiles (Fig. 1). There was a slight bias toward higher %G+C for the NRs and lower %G+C in the CRs, which could reflect a bias in the assembly algorithm.

Tetranucleotide frequency has been shown to be capable of distinguishing higher taxonomic classifications, up to the species level (*Teeling et al., 2004*; *Dick et al., 2009*). This resolving power has been leveraged in binning protocols (*Tyson et al., 2004*; *Wu, Simmons & Singer, 2016*; *Albertsen et al., 2013*; *Imelfort et al., 2014*). To investigate whether genomic regions with divergent TNF are poorly recovered in genome reconstruction, we compared the TNFs of CRs and NRs to that of the cognate complete genome using chi-squared analysis. In most cases, the chi-squared statistic was an order of magnitude higher for NRs vs CRs, and the differences were significant for all chromosomal sequences except for HL-46, HL-109, HL-93 and the small chromosome of HL-91 (Table 3).

One factor that could affect nucleotide composition effects on binning is the length of the region with divergent composition vs the length of the contig. If the variant region comprises most of the length of the contig being evaluated, the difference from the genome average will be pronounced, whereas if the divergent region is only a small percentage of the contig length, the signal will be muted. An examination of CR/NR length vs compositional variance (Fig. S3) revealed a strong, significant negative correlation between contig length and TNF chi square for CRs ($R^2 = 0.64$, *p*-value $< 2.2 \times 10^{-16}$) and a weaker relationship for NRs ($R^2 = 0.14$, *p*-value $= 4.9 \times 10^{-12}$). Taken together, the %G+C and

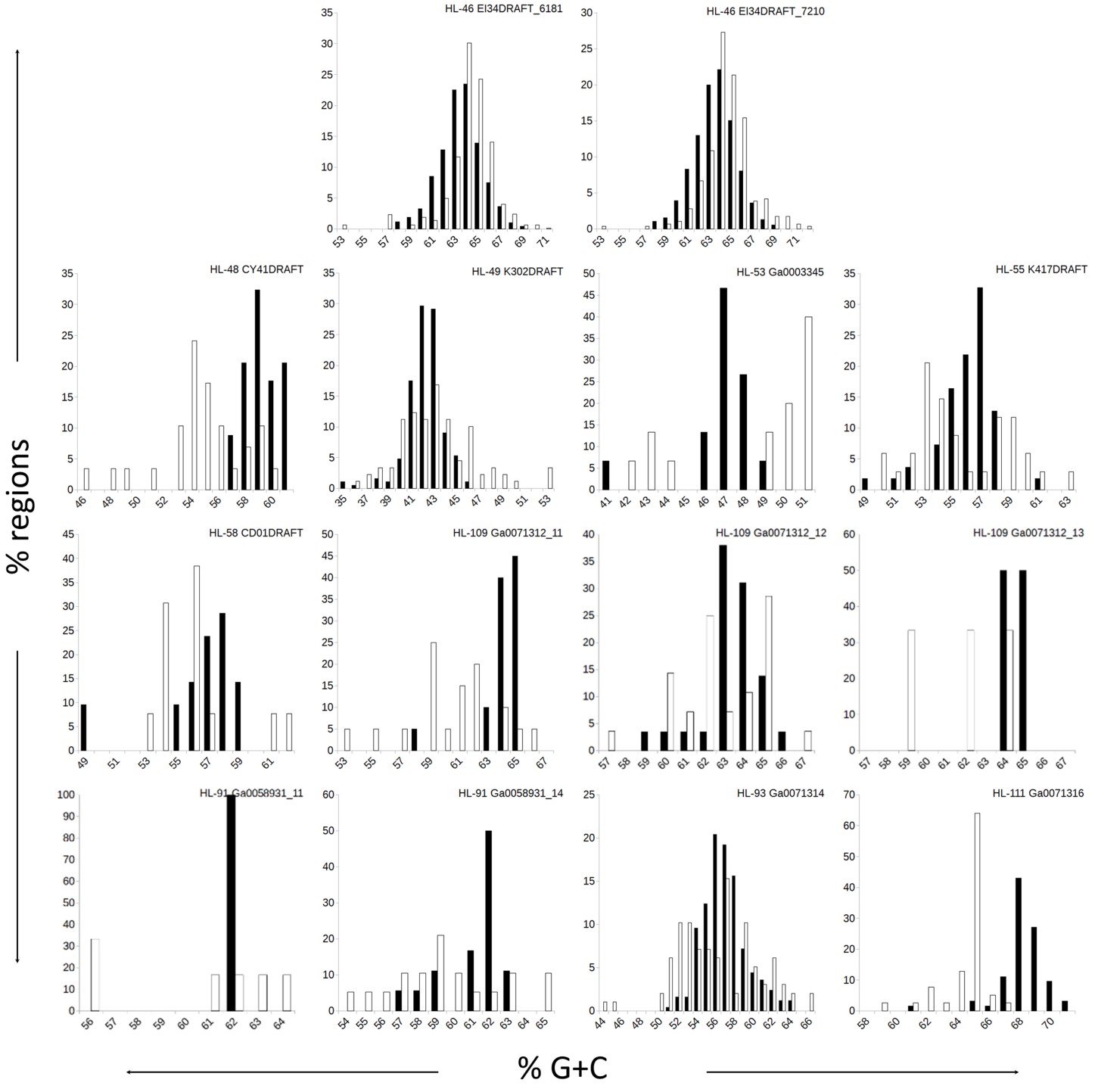

**Figure 1 Distributions of %G+C for MDR and CDR genomic regions.** G+C composition was determined for individual regions identified as CDRs or MDRs. Bar height represents the percentage of regions in the category. Black bars, CDRs; white bars, MDRs.

TNF results show that genomic regions with divergent nucleotide composition are more likely to be missed during binning, and this effect is stronger for short contigs. The most effective way to overcome this problem is to enhance assembly such that regions

**Table 3 Tetranucleotide frequency analysis.**

|  | Molecule | CR | | | NR | | |
|---|---|---|---|---|---|---|---|
|  |  | Mean | sd | p-value | Mean | sd | p-value |
| HL-46 | EI34DRAFT_6181 | 0.2323 | 0.1883 | 0.154 | 0.1518 | 0.1429 | 0.983 |
| HL-46 | EI34DRAFT_7210 | 0.2042 | 0.0696 | 0.896 | 0.1701 | 0.1332 | 0.975 |
| HL-48 | CY41DRAFT | 0.0276 | 0.0577 | 0.387 | 0.4425 | 0.2689 | **<0.001**[a] |
| HL-49 | K302DRAFT | 0.0522 | 0.0431 | 0.757 | 0.2340 | 0.2164 | **<0.001** |
| HL-53 | Ga0003345 | 0.0261 | 0.0451 | **0.001** | 0.3851 | 0.1525 | **<0.001** |
| HL-55 | K417DRAFT | 0.0458 | 0.0726 | 0.086 | 0.2774 | 0.2168 | **0.004** |
| HL-58 | CD01DRAFT | 0.0761 | 0.1451 | 0.008 | 0.2974 | 0.969 | **0.004** |
| HL-91 | Ga0058931_11 | 0.0266 | 0.0213 | 0.313 | 0.3043 | 0.1416 | 0.011 |
| HL-91 | Ga0058931_12 | nd[b] | nd | nd | nd | nd | nd |
| HL-91 | Ga0058931_13 | nd | nd | nd | nd | nd | nd |
| HL-91 | Ga0058931_14 | 0.0557 | 0.0647 | **0.004** | 0.3614 | 0.2052 | **<0.001** |
| HL-93 | Ga0071314_11 | 0.0925 | 0.0738 | 0.993 | 0.2254 | 0.1595 | 0.062 |
| HL-109 | Ga0071312_11 | 0.0262 | 0.0401 | 0.396 | 0.3148 | 0.1842 | 0.087 |
| HL-109 | Ga0071312_12 | 0.0216 | 0.0281 | 0.076 | 0.2907 | 0.1913 | 0.231 |
| HL-109 | Ga0071312_13 | 0.0048 | 0.0019 | 0.538 | 0.3651 | 0.2299 | 0.016 |
| HL-111 | Ga0071316_11 | 0.0396 | 0.0561 | 0.322 | 0.4504 | 0.1640 | **<0.001** |

**Notes:**
[a] Bold text indicates significant result.
[b] Not determined because the entire molecule was missing from the reconstructed genome.

with unusual content are included in significantly longer contigs, or use clone linkage to identify strong, unique connections to binned contigs.

## Repeated sequences segregate aberrantly

Sequence coverage profiles are frequently effective in discriminating contigs from different organisms (*Tyson et al., 2004*). Samples taken under different conditions or at different times capture community states which have similar organismal composition but differing relative abundances. This difference translates to distinct coverage profiles for assembled contigs, and thus contigs with similar coverage profiles are assumed to originate from the same organism. In this data set, for example, we compared two cultures with near-identical heterotroph species composition, but different cyanobacteria acting as a conduit for energy and carbon (*Cole et al., 2014*; *Nelson et al., 2015*). Other studies have compared samples taken at different times (*Albertsen et al., 2013*). Coverage analysis is more difficult for repeated regions of a genome, which will yield higher coverage values than the genome average and thus are more likely to be either not binned or binned improperly. Differential coverage analysis can mitigate this problem by identifying correlated changes in abundance of contigs with different coverage. Unlike nucleotide composition variance; however, unusual high-coverage signal due to repeat sequence is less likely to be diluted by incorporation into a larger contig because assemblers (especially standard de Bruijn graph assemblers using short-read data) tend to terminate contigs when repeats are encountered and/or assemble repeats into separate contigs (*Pop, 2009*).

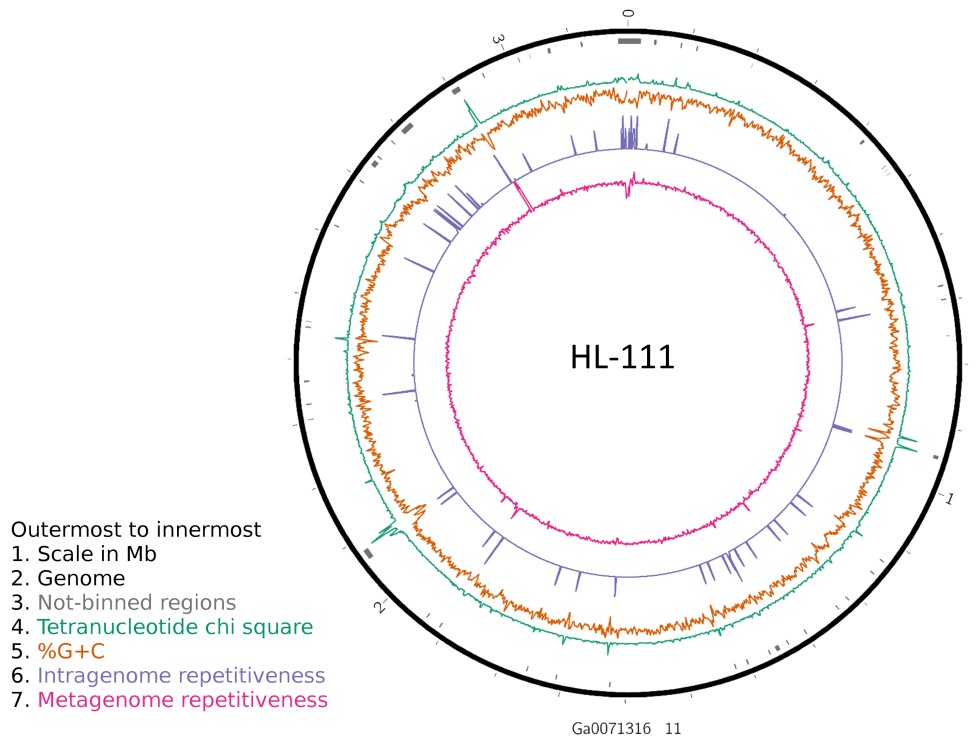

Outermost to innermost
1. Scale in Mb
2. Genome
3. Not-binned regions
4. Tetranucleotide chi square
5. %G+C
6. Intragenome repetitiveness
7. Metagenome repetitiveness

Ga0071316 11

**Figure 2** **Analysis of HL-111 genome.** Ring 1 (outermost, black), genome sequence; ring 2 (grey bars), missed detection regions (MDRs); ring 3 (teal), tetranucleotide frequency (TNF) distance $\chi^2$ values; ring 4 (orange), %G+C; ring 5 (blue), intragenome redundancy; ring 6 (magenta), metagenome redundancy. Values were calculated across 2,000 nt windows with a step size of 1,000 nt. For TNF, $\chi^2$ was calculated for the windows using the whole molecule frequencies as the expected. Data for other genomes analyzed is presented in Fig. S1. Circlular plots were generated using Circos v0.69.3 (*Krzywinski et al., 2009*).

To examine the impact of repeated sequences on genome reconstruction, we determined the repetitiveness of sequence information across CRs and NRs, determined from a self-vs-self similarity search, and compared those values to the genome average. Correspondence of repeated regions and NRs was strong (Figs. 2 and 3; Fig. S4). In HL-111, all NRs save one were present in at least two copies (Fig. 2). For all reconstructions, save HL-46, the CRs had repeat content equal to or lower than the genome average.

Another phenomenon that can affect contig coverage in metagenomic assembly is multiple organisms sharing identical regions of DNA. Some regions are highly conserved between related species, an example being the ribosomal RNA operon, which is known to confound assemblers and segregation strategies (*Ghurye, Cepeda-Espinoza & Pop, 2016*). Alternatively, mobile elements such as plasmids or transposons can have a broad host range and invade and inhabit closely or even distantly related organisms (*Frost et al., 2005*). Such regions, even if not repeated within a genome, will exhibit anomalous coverage and thus could be either excluded or mis-binned. We examined the metagenomic read coverage depth to determine if NRs had anomalous profiles relative to the whole genome and the CRs. For most reconstructions, the coverage of NRs differed

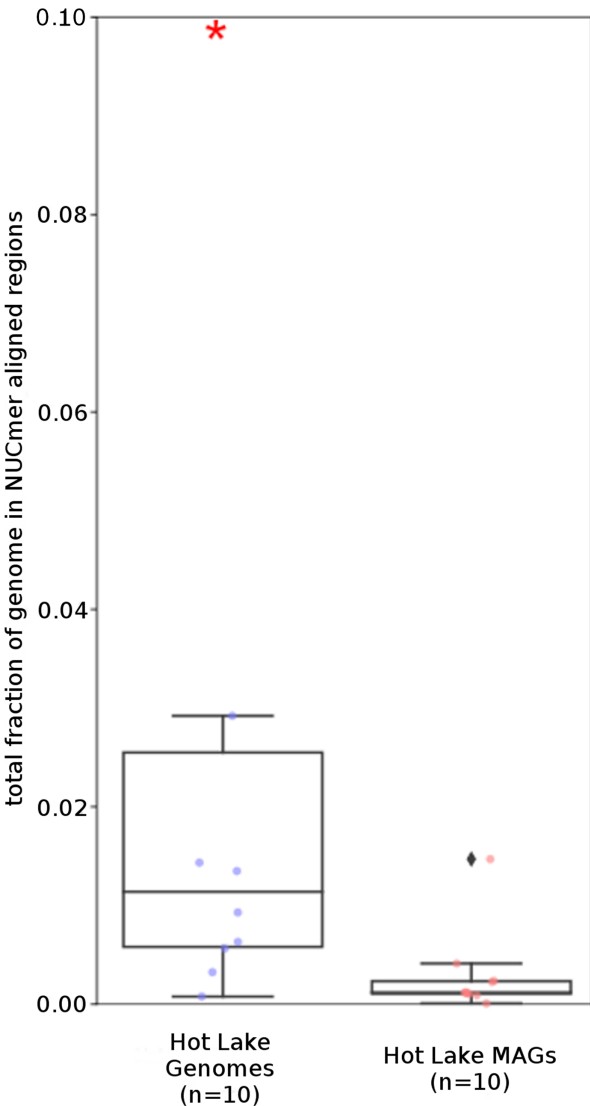

**Figure 3 Repeat content of genomes vs MAGs.** Box plot representation of the total fraction of each genome/MAG in a repeat region as determined by NUCmer (≥97% identity; center line, median; box limits, upper and lower quartiles; whiskers, 1.5× interquartile range; diamonds, outliers). UCC MAG and genome comparison were significantly different ($p = 0.01$; Mann–Whitney U). Red asterisks denote the existence of outliers outside of the displayed dataset.

from the genome average and that of the CRs (Table 4; Fig. 2; Fig. S4). Only HL-46 and one of the HL-109 molecules did not have significant differences. Most NRs displayed higher or equivalent coverage values, however, several NRs in HL-48 and the two small plasmids associated with HL-91 showed lower metagenomic coverage values (Fig. S4). A likely explanation for this is the presence in the consortia of sub-populations of these organisms that lack the plasmids.

## Functional assessment of NR genes

To determine the extent to which regions missing from reconstructions might affect downstream metabolic or functional analyses and predictions for organisms and

**Table 4 Genomic redundancy.**

| | Molecule | Genome | CR | | | NR | | |
|---|---|---|---|---|---|---|---|---|
| | | Mean | Mean | Distance | *p*-value | Mean | Distance | *p*-value |
| HL-46 | EI34DRAFT_6181 | 2.76 | 2.78 | 0.26 ± 0.15 | 0.992 | 2.42 | 0.43 ± 0.31 | 0.264 |
| HL-46 | EI34DRAFT_7210 | 5.98 | 4.43 | 2.95 ± 2.34 | 0.978 | 4.99 | 4.01 ± 13.35 | 0.649 |
| HL-48 | CY41DRAFT | 72.40 | 69.29 | 3.65 ± 2.45 | 1.000 | 140.93 | 100.97 ± 153.33 | **<0.001**[a] |
| HL-49 | K302DRAFT | 8.97 | 8.67 | 0.51 ± 0.58 | 0.999 | 11.38 | 4.16 ± 17.52 | **0.002** |
| HL-53 | Ga0003345 | 441.81 | 446.29 | 24.51 ± 18.21 | 0.073 | 517.07 | 115.56 ± 59.71 | **<0.001** |
| HL-55 | K417DRAFT | 16.76 | 15.35 | 7.06 ± 10.45 | 0.679 | 117.37 | 110.35 ± 333.81 | **<0.001** |
| HL-58 | CD01DRAFT | 128.28 | 127.85 | 9.10 ± 15.71 | 1.000 | 180.14 | 60.44 ± 27.54 | **<0.001** |
| HL-91 | Ga0058931_11 | 231.39 | 228.46 | 3.64 ± 2.25 | 0.786 | 311.6 | 91.27 ± 97.44 | **0.001** |
| HL-91 | Ga0058931_12 | 163.24 | nd[b] | nd | nd | nd | nd | nd |
| HL-91 | Ga0058931_13 | 168.27 | nd | nd | nd | nd | nd | nd |
| HL-91 | Ga0058931_14 | 227.56 | 231.77 | 8.18 ± 6.59 | 0.220 | 273.03 | 97.82 ± 117.47 | **<0.001** |
| HL-93 | Ga0071314_11 | 50.87 | 50.03 | 4.04 ± 2.92 | 1.000 | 65.73 | 16.16 ± 35.87 | **<0.001** |
| HL-109 | Ga0071312_11 | 3103.11 | 3098.73 | 97.47 ± 72.15 | 0.748 | 3072.59 | 323.24 ± 240.86 | **0.005** |
| HL-109 | Ga0071312_12 | 2821.18 | 2822.26 | 113.08 ± 78.18 | 0.124 | 2778.03 | 352.81 ± 436.28 | **0.003** |
| HL-109 | Ga0071312_13 | 2853.84 | 2901.40 | 47.56 ± 9.73 | 0.179 | 2097.01 | 756.83 ± 256.91 | 0.018 |
| HL-111 | Ga0071316_11 | 90.14 | 88.03 | 3.98 ± 4.31 | 0.993 | 98.25 | 38.42 ± 104.87 | 0.027 |

**Notes:**
[a] Bold text indicates significant result.
[b] Not determined because the entire molecule was missing from the reconstructed genome.

communities, we examined the gene content of the NRs and the functional roles of those genes. COG categorization was used as a basis for comparison because of its ability to identify, in particular, genes associated with mobile elements such as plasmids, phage and insertion sequences. In addition, we evaluated the distribution of non-coding RNA genes since some are known to be repeated within genomes (multiple rRNA operons, for example), and others (tRNAs) are commonly associated with mobile elements (*Hacker & Kaper, 2000*).

For all the reconstructions, the gene content of the NRs differed from that of the CRs and complete genomes. Functional analysis of gene sequences shows that this difference was largely driven by genes encoding mobile element functions (COG category X) and RNA genes (Fig. 4). The mobile element genes in the NR regions were predominantly transposases with some contribution from bacteriophage and plasmid genes (HL-91; HL-93). Most of the identified rRNA genes fell within NRs, with only HL-48 and HL-53 each having one rRNA contained in a CR. In addition, the NRs, including the two entire plasmids from HL-91 which were not binned, contained a higher percentage of genes that were not assigned to a COG category.

## Evaluation of a complex metagenomic data set and common automated binning tools

To verify that our conclusions of genome reconstruction bias in the highly curated UCC data set were extendable to more complex data sets and for alternate, widely-used binning

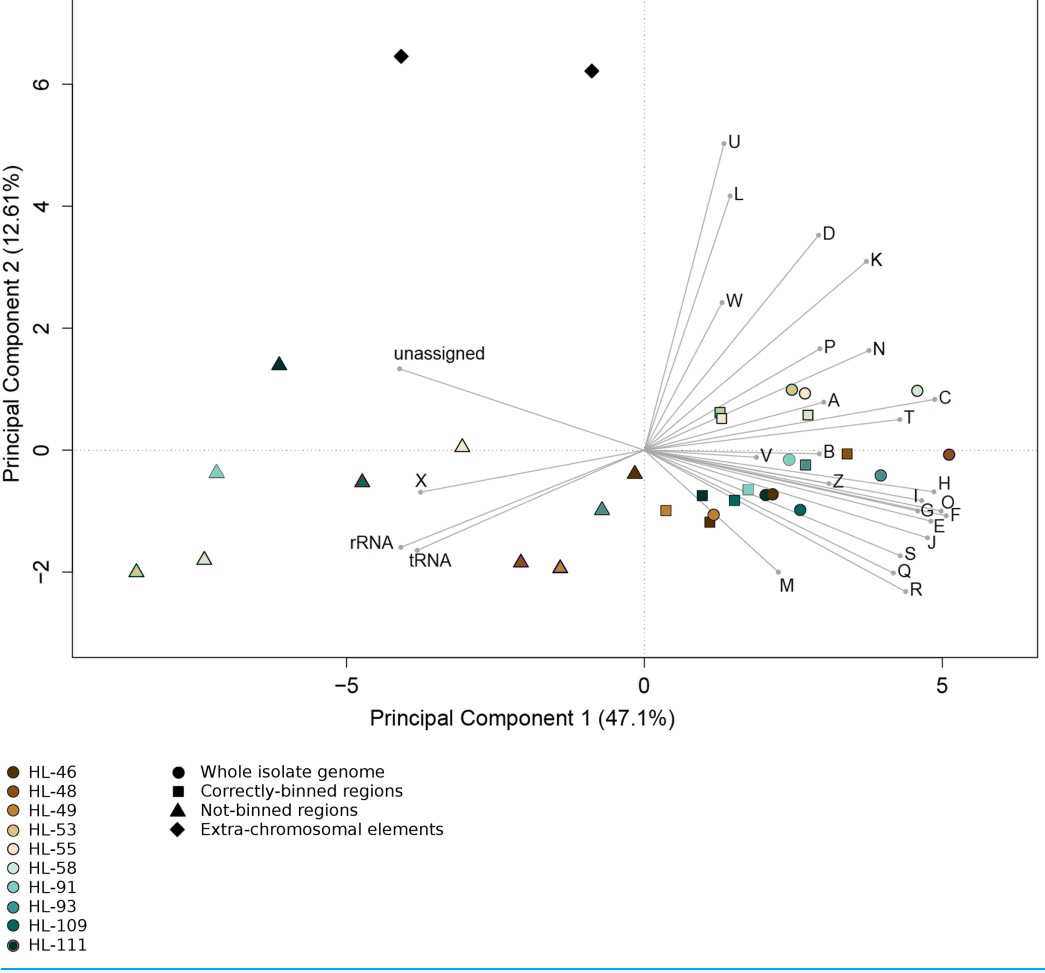

**Figure 4 Functional categorization of genes present on MDRs.** The gene features of each genome region were assigned to functional COG categories or as non-coding genes (rRNA; tRNA; ncRNA). Organisms' gene sets were compared using Principal Component Analysis. Organisms are represented by colors (HL-46, yellow; HL-48, purple; HL-49, blue; HL-53, light blue; HL-55, gray; HL-58, orange; HL-91, black; HL-93, pink; HL-109, red; HL-111, green). The genome region categories are represented by shapes (whole isolate genomes, circles; CDRs, squares; MDRs, triangles; extrachromosomal elements, diamonds). COG categories: *A*, RNA processing and modification; *B*, Chromatic structure and dynamics; *C*, Energy production and conversion; *D*, Cell cycle control, cell division, chromosome partitioning; *E*, Amino acid transport and metabolism; *F*, Nucleotide transport and metabolism; *G*, Carbohydrate transport and metabolism; *H*, Coenzyme transport and metabolism; *I*, Lipid transport and metabolism; *J*, Translation, ribosomal structure and biogenesis; *K*, Transcription; *L*, DNA replication, recombination and repair; *M*, Cell wall/membrane/envelope biogenesis; *N*, Cell motility; *O*, Post-translational modification, protein turnover, chaperones; *P*, Inorganic ion transport and metabolism; *Q*, Secondary metabolites biosynthesis, transport and catabolism; *R*, General function prediction; *S*, Function unknown; *T*, Signal transduction mechanisms; *U*, Intracellular trafficking, secretion and vesicular transport; *V*, Defense mechanisms; *W*, Extracellular structures; *X*, Mobilome, transposons, phages; *Y*, Nuclear structure; *Z*, Cytoskeleton.

tools, we applied similar analyses to MAGs generated from the *Tara* Oceans metagenomic data using distinct genome reconstruction protocols. For this comparison, 4,557 MAGs generated from the *Tara* Oceans microbial metagenomic data reconstructed using three complementary methods were collected and analyzed. Three different automated

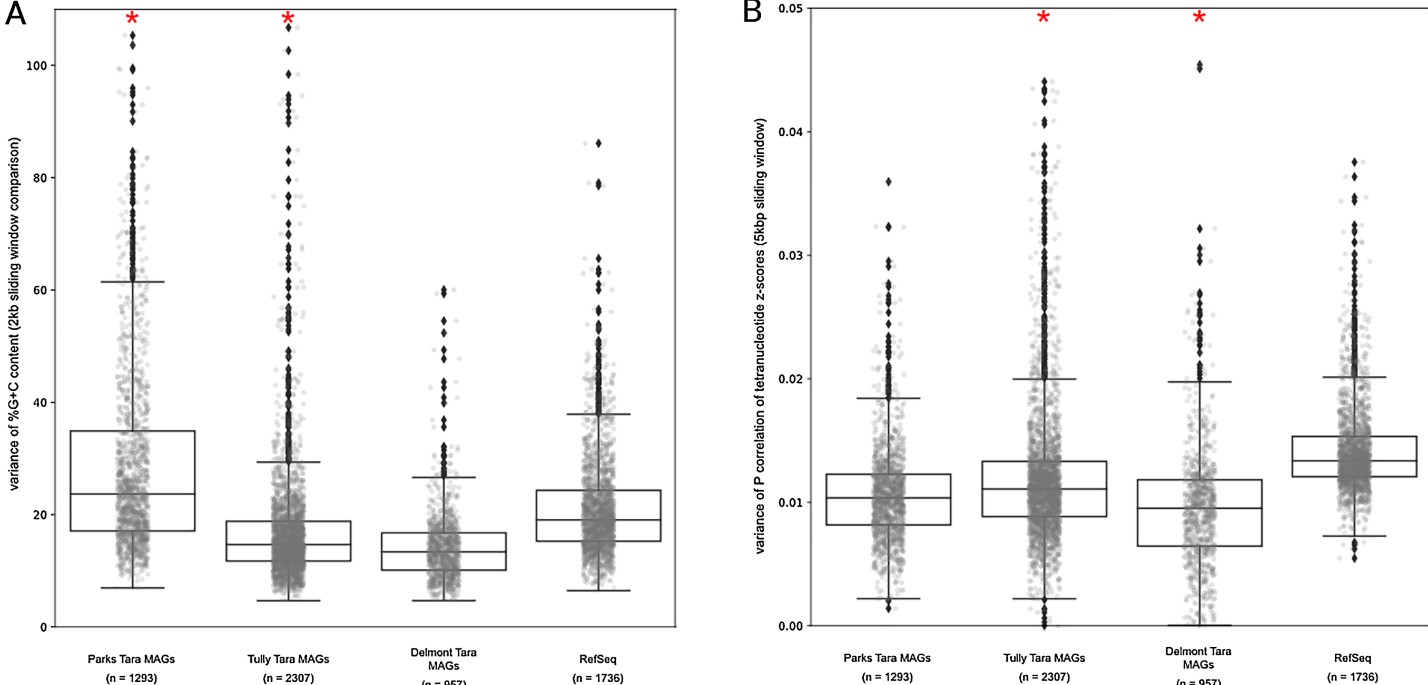

**Figure 5** *Tara* **Ocean MAG nucleotide composition analysis.** (A) %G+C variance analysis. Box plot representation of the %G+C variance for each 2,000 bp segment of genome/MAG (sliding window step: 500 bp; center line, median; box limits, upper and lower quartiles; whiskers, 1.5× interquartile range; diamonds, outliers). Comparisons between *Tara* Oceans MAG datasets and RefSeq genomes were significantly different ($p < 0.001$; Mann–Whitney U with Benjamini-Hochberg False Discovery Rate Correction (BH FDR)). (B) Tetranucleotide analysis. Box plot representaiton of the variance in Pearson correlation values of the tetranucleotide *Z*-scores for a pair-wise comparison of each 10 kb segment of genome/MAG (sliding window step: 5 kb; center line, median; box limits, upper and lower quartiles; whiskers 1.5× interquartile range; diamonds, outliers). Comparisons between *Tara* Oceans MAG datasets and RefSeq genomes were significantly different ($p < 0.001$; Mann–Whitney U with BH FDR Correction). Red asterisks denote the existence of outliers outside of the displayed range.

binning methodologies were employed to generate the MAG data set: MetaBat (v0.26.3) (*Parks et al., 2018*; *Kang et al., 2015*), BinSanity (v1.0) (*Tully, Graham & Heidelberg, 2018*; *Graham, Heidelberg & Tully, 2017*), and CONCOCT (with manual refinement in anvi'o) (*Eren et al., 2015*; *Delmont et al., 2018*). All three automated binning algorithms utilized read coverage and TNF to identify congruent contigs, with the intended role of the algorithms to reconstruct high confidence environmental genomes while avoiding over-binning (i.e., removing elements that deviate from the mean values of the binned contigs). The MAGs had a mean estimated completeness and contamination of 76.6% and 2.2%, respectively, as determined by CheckM v.1.1.1 (*Parks et al., 2015*). In comparison, 1,736 "representative" and "reference" complete genomes were collected from NCBI RefSeq.

Our results above predicted that the MAGs would have lower %G+C variance and TNF variance than the isolate complete genome data set. For the observed %G+C, MAGs tended to have lower variance ($p < 0.001$) than isolate genomes (Fig. 5A). The exception was the *Parks et al. (2018)* MAGs, which had a much larger variance, even compared to the RefSeq genome set (mean vs mean, $p < 0.001$). This may be the result of the additional step applied to the MAGs by *Parks et al. (2018)*, whereby related MAGs with

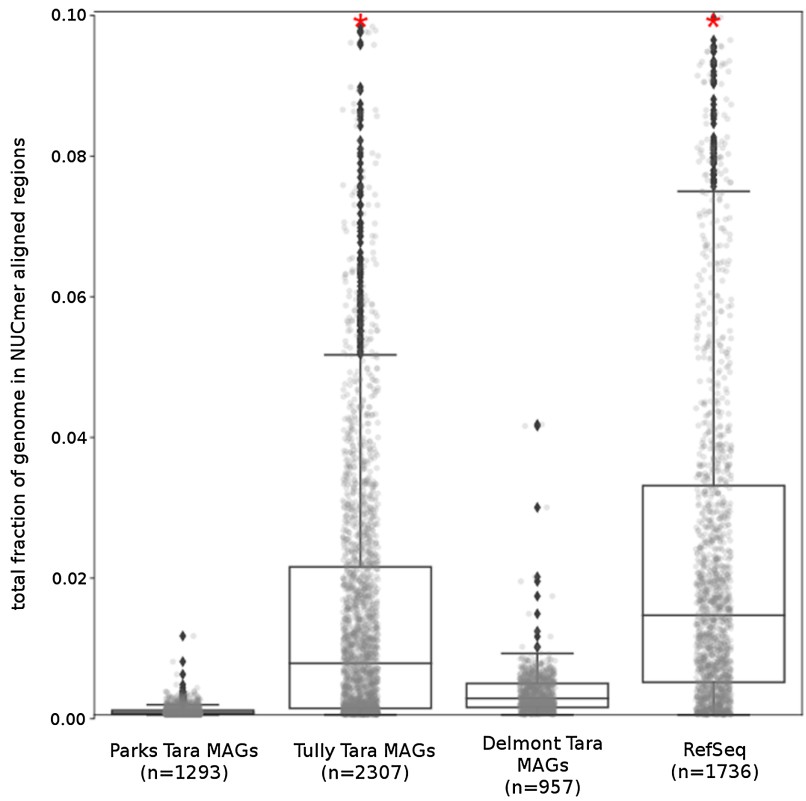

**Figure 6 *Tara* Ocean MAG *repeat* content.** Box plot representation of the total fraction of each genome/MAG in a repeat region as determined by NUCmer (≥97% identity; center line, median; box limits, upper and lower quartiles; whiskers, 1.5× interquartile range; diamonds, outliers). Comparisons between *Tara* Oceans MAG datasets and RefSeq genomes were significantly different ($p < 0.001$; Mann–Whitney U with BH FDR Correction). Red asterisks denote the existence of outliers outside of the displayed dataset.

<3% mean %G+C difference were merged into a single representative MAG. For the *Tully, Graham & Heidelberg (2018)* and *Delmont et al. (2018)* MAGs, the lower variance observed compared to the RefSeq genomes is likely due to removal of contigs with deviant %G+C values during binning. The MAGs also had lower variance with regards to TNF compared to the RefSeq genomes ($p < 0.001$) (Fig. 5B), again, likely due to genomic elements that deviated from the average value of the binned contigs having been removed during the binning steps. These observations support our conclusions regarding genome regions having divergent nucleotide composition being underrepresented in MAGs.

The *Tara* and NCBI Refseq data sets were then evaluated for repeat sequence content. Each MAG and isolate genome was compared to itself using NUCmer to identify the fraction of the genome composed of repeat regions (regions with ≥97% sequence identity). MAGs universally had a smaller fraction of genomic information in repeat regions compared to isolate genomes ($p < 0.01$; Fig. 6). The lack of repeat regions in MAGs is likely the result of repeated regions having inflated or depressed read coverage values relative to the mean of the genome, depending on the number of copies of the repeat region present in the genome and how stable this number is across the population. Compared to

the other *Tara* MAGs, the *Tully, Graham & Heidelberg (2018)* MAGs had a larger fraction of redundant genomic elements. It is unclear what aspect of the assembly and binning methodology has influenced these results. On average, the lengths of the repeat regions from the *Tully, Graham & Heidelberg (2018)* MAGs are longer than the repeat regions in the RefSeq genomes (mean: 1,052 bp vs 868 bp, respectively).

## What's missing from reconstructed genomes?

Analysis of regions that were not recovered from genome reconstruction (NRs) showed both nucleotide compositional variance and intragenome repetitiveness. The %G+C and tetranucleotide frequencies of NRs tended to differ from that of complete genomes (Tables 2 and 3; Fig. 1), and the sequence coverage differed. This met expectations since, in general, binning tools are designed around the assumption that sequences with similar properties belong together, thus any genome region that varies significantly from the genome average is likely to be incorrectly binned if it comprises the majority of a contig under consideration. Regions with atypical nucleotide content have been observed to contain genes upon which selective pressures are acting on nucleic acid structure, such as ribosomal RNAs and tRNAs (*Galtier & Lobry, 1997*; *Hurst & Merchant, 2001*; *Schattner, 2002*), and exogenously introduced segments such as mobile elements (*Daubin, Lerat & Perriere, 2003*; *Garcia-Vallve, Romeu & Palau, 2000*). It is significant that many of the NRs displayed lower %G+C than the genome average, since it has been observed that laterally acquired regions tend to have lower %G+C than their hosts (*Daubin, Lerat & Perriere, 2003*), as phage and insertion sequences tend to have A+T-enriched genomes (*Rocha & Danchin, 2002*). Notably, many genome regions with variant nucleotide composition were incorporated into longer contigs by the assembler, masking the variance and allowing correct binning. Conversely, the assembler collapsed repeated region sequences into single contigs, and thus they were not binned due to the inflated sequence coverage values. Often, repeated sequences displayed divergent nucleotide composition, but the reciprocal was less frequent, indicating that repetitiveness is the stronger driver of binning failure. These results demonstrate that assembly efficiency is an important determining factor for correct binning, or conversely, any factor that results in shorter assemblies will result in poorer recovery of anomalous regions. Thus, it is advisable to include replication and positive controls in metagenomic sequencing protocols, particularly for highly diverse communities such as soils and riverbed sediments, to allow evaluation of assembly efficiency and accuracy.

Repeat regions identified in this study appeared to largely consist of insertion elements based on functional analysis and their relatively short size (1–2 kb). Failure of these regions to be correctly binned is unlikely to meaningfully affect functional predictions for a reconstructed genome. Their presence in a genome is more likely to affect metabolic reconstruction analysis by reducing assembly efficiency, resulting in more, shorter contigs and increasing the chance that these shorter contigs are not binned or incorrectly binned. Technological advances increasing read length beyond two kb will increase contig lengths, binning accuracy, and the likelihood of yielding closed genomes from environmental samples (*White et al., 2016*; *Frank et al., 2016*; *Bertrand et al., 2019*).

NRs were generally observed to be short, with a median length of less than five kb (Table 1) and containing only a handful of genes. Thus, even a MAG with many gaps (indicating a large number of NRs) may be missing only a small percentage of its genome. The conserved single-copy gene (CSCG) estimations for completeness appear for all intents and purposes to be a reasonable indication of how much information is absent (*Nelson et al., 2015*). One caveat to this conclusion, however, is that extrachromosomal elements, plasmids and phages (integrated or otherwise) typically do not carry CSCG markers, and thus are essentially invisible in such analyses. The longer NRs observed in our analysis appear to comprise integrated plasmids or phage, and thus any gap in a reconstruction could represent up to 50 kb (or more) of genetic material. Importantly, these represent introduced genetic material, which, while likely conveying a beneficial trait, are unlikely to carry functions that are integral to host metabolic function.

## CONCLUSIONS

This analysis indicates that reconstructed genomes estimated to be near-complete can be assumed to contain nearly all genes important to metabolic reconstruction. The majority of identifiable genes present on NRs appear to be either highly conserved, non-coding genes that can be assumed to be present (such as the rRNA genes and tRNA genes) or are associated with mobile genetic elements. While many of these genes may be not be directly related to cellular metabolism (transposases, toxin/antitoxin systems, phage and plasmid functions), it should be noted that entire extrachromosomal elements may be missed by the binning process due to either alternate nucleotide composition, a higher number of copies per cell than the genome, or occupancy in only a subset of the population (such as the two molecules in HL-109). These elements frequently carry genes that alter the physiology or resistance of the host organism. For example, HL-109 and HL-111 have NRs that includes genes involved in glycan biosynthesis, suggesting alterations to the cell wall, while HL-91 has picked up a multidrug efflux transporter. As such, reconstructed genomes can be considered reliable foundations for metabolic reconstruction but should not be assumed to be comprehensive for the function of the organism.

## ACKNOWLEDGEMENTS

The authors would like to thank Jim Fredrickson and Lori Nelson for critical evaluation of the manuscript during preparation. This is C-DEBI Contribution 550.

### Funding

William C. Nelson and Jennifer M. Mobberley were supported by the U.S. Department of Energy (DOE), Office of Biological and Environmental Research (BER), as part of BER's Genomic Science Program (GSP). This contribution originates from the GSP Foundational Scientific Focus Area (FSFA) at the Pacific Northwest National Laboratory (PNNL). The Pacific Northwest National Laboratory is operated for DOE by Battelle Memorial

Institute under contract DE-AC05-76RL01830. Sequence data presented was generated at the DOE Joint Genome Institute under contract no. DE-AC02-05CH11231 and Community Science Project 701. Benjamin J. Tully was funded through the Center for Dark Energy Biosphere Investigations (OCE-0939654). The funders had no role in study design, data collection and analysis, decision to publish, or preparation of the manuscript.

### Grant Disclosures

The following grant information was disclosed by the authors:
U.S. Department of Energy (DOE).
Office of Biological and Environmental Research (BER).
BER's Genomic Science Program (GSP).
Pacific Northwest National Laboratory (PNNL).
Battelle Memorial Institute: DE-AC05-76RL01830.
DOE Joint Genome Institute: DE-AC02-05CH11231.
Community Science Project 701.
Center for Dark Energy Biosphere Investigations: OCE-0939654.

### Competing Interests

The authors declare that they have no competing interests.

### Author Contributions

- William C. Nelson conceived and designed the experiments, performed the experiments, analyzed the data, prepared figures and/or tables, authored or reviewed drafts of the paper, and approved the final draft.
- Benjamin J. Tully conceived and designed the experiments, performed the experiments, analyzed the data, prepared figures and/or tables, and approved the final draft.
- Jennifer M. Mobberley conceived and designed the experiments, performed the experiments, analyzed the data, prepared figures and/or tables, authored or reviewed drafts of the paper, and approved the final draft.

### Data Availability

The raw metagenomic data used to construct the MAGs is available in the NCBI Sequence Read Archive: SRX1063989 and SRX1065184.

The Tara Oceans MAG data is available as described in Table S1, and in the original publications: Tully et al. (DOI 10.1038/sdata.2017.203), Parks et al. (DOI 10.1038/s41564-017-0012-7) and Delmont et al. (DOI 10.1038/s41564-018-0176-9).

The NCBI RefSeq genomes used in the analysis are available in Table S2.

All custom analysis scripts are available at https://github.com/wichne/biases_in_genome_reconstruction.

The Hot Lake unicyanobacterial consortia MAG and genome data analyzed are available in GenBank (Table 1).

## Supplemental Information

Supplemental information for this article can be found online at http://dx.doi.org/10.7717/peerj.10119#supplemental-information.

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
