# Peer review of "Biases in genome reconstruction from metagenomic data"

_PeerJ, doi:10.7717/peerj.10119_

## Round 0.1 · original submission · Major Revisions

The manuscript was in general well written; however, the experimental design was too narrow in that it focused on results from only one metagenomic assembly. The manuscript deals with an important topic, and to gain acceptance from the readers would require a more thorough observation of the suggested approach by testing different kinds of populations and using a variety of software applications which are used for the assembly process. A single discussion of one study helps the reader understand the interpretation; however, does not provide evidence of its utility for other studies or conditions. The general structure of the manuscript is a bit unusual in that some references from the introduction are repetitively introduced in the results and discussion sections. As provided by the reviewers the general scope of the manuscript would benefit from expanding the study to provide some sort of trend with additional data to be convincing for the presented protocol. The observations are generally known, and the attempts provided here appear valid; but, without other studies included are not convincing for success. Thus, a major revision is suggested. Perhaps this is a beginning of a review for difficulties in assembly and does fall short in some respects for that. If you would address comments from the reviewers and perhaps provide additional data, this would help improve receptivity of the manuscript. Thank you for your contribution and we look forward to a modified response.

Reviewer 1 ·

Basic reporting

This manuscript described the biases in the genome reconstruction process from metagenomic data. The authors explored several factors that might affect the genome reconstruction by comparing their unicyanobacterial consortia paralleled genomic and metagenomic datasets. They reported that repeated sequences and genomic regions with variant nucleotide compositions are more likely to be missing, and that missing regions are strong biased toward rRNA, tRNA, and mobile elements.

Although most of the conclusions drawn from this study are not very surprising, the study itself is plausible in that it dig through the elements that might affect the genome reconstruction process. There are however still room to improve this study. I give my review as follows.

1. The dataset that the authors used for probing the problem is the unicyanobacterial consortia that they developed as model systems, which however represents just one system. This makes their conclusions a bit weaker since their results were only drawn from that very system. I would appreciate if the authors can extend their study to one or two more systems, be them simulated or real ones with genuine answers.

2. The microbial population complexity may also affect the genome reconstruction results. Looks like the unicyanobacterial consortia is a simple system since not too many genomes can be recovered and reconstructed. Perhaps the study can be extended to a more complex microbial community. The authors may also be able to say something about the relationships between microbial community complexity and the reconstruction process.

3. I wonder when the authors term MDR (missed detection region), do they distinguish contigs that were not binned correctly between contigs that were filtered out due to short contig sizes? For example, 16S rRNA genes cannot usually be assembled very well and were usually split into pieces. Hence these fragmented were usually filtered out in the very first step of binning due to the contig length requirements of most binning tools. This also applies to most repetitive elements since they confuse assemblers. I guess the analysis of MDR is focused on those long enough to be included in the binning process and would very much appreciate if the short contigs can also be included in the analysis as well.

4. (line 109) how come tetranucleotide frequencies need to be extracted across all six reading frames? Tetranucleotide need to be considered for only forward and reverse-complement strains. Please explain or revise this part.

5. In table 3 there is a “mean tetranucleotide frequency,” however I do not know what does that mean. Are all tetranuclotide frequencies for the CDR or MDR regions of organisms be calculated and averaged? Perhaps the authors can explain more in both table 3 and the text.

6. Two words are used interchangeably in this manuscript: repeat and redundancy. I however feel that the word “redundancy” is a bit misleading and was not used very frequently throughout genomic analysis. Alternatively the term “per-base coverage” may be more informative. Please consider using terms that are more widely used and comprehensible.

7. Figure 2 is quite clear in delivering the message to readers. I wonder if the values such as TNF, GC%, or “redundancy” is derived by a sliding window?

Experimental design

no comment

Validity of the findings

no comment

Additional comments

no comment

·

Basic reporting

In the manuscript entitled ‘Biases in genome reconstruction from metagenomic data’, authors describe genomic regions associated with ten microbial populations that were missing in metagenome-assembled genomes, yet present in culture representatives. Authors found that the missing regions exhibit distinct traits compared to the rest of the genomes studies, and conclude that high-completion metagenome-assembled genomes are generally good representatives of the metabolic potential of the microbial populations they represent.

I note that links to the genomic and metagenomic raw reads used for mapping are not available and need to be added in the material and methods section.

Experimental design

The experimental design used in this manuscript is valid. Authors applied the same bioinformatics methodology to genomic regions present and missing in metagenome-assembled genomes (GC%, sequence composition, coverage, and functional potential). Methods are described with enough details.

I would like to mention one limit of the experimental design. There are many ways to characterize metagenome-assembled genomes, and it would have strengthened to study to test the effect of assembly (using different software) and binning (using both manual and automatic tools) on downstream results. Are the missing regions due to fundamental limitations of assembly and/or binning, or specific to the tools used to characterize the metagenome-assembled genomes? To me, this is the main limitation of the study. However, it does not impact the described observations. In my opinion, it merely limits the extent of the conclusions and does not prevent publication in its current form.

Validity of the findings

The findings of this study are valid, and in line with the field’s understanding of assembly-based metagenomics.

Additional comments

The overall aim of the study is to investigate the genomic regions of microbial populations systematically missing when using assembly-based metagenomics, as compared to cultivation. This is an important topic, as metagenome-assembled genomes largely contribute to our understanding of the microbial tree of life. Authors demonstrated that in the studied microbial community, nucleotide composition of genomic regions missing in reconstructed “genomes from metagenomes” frequently differ from the genome average. This contributes to our understanding of the limits of metagenomic assemblies, and/or binning. The manuscript is well written, and general trends appear to emerge from the analysis of 10 microbial populations.

The manuscript is of interest, and I could not see any flaw in the methodology. However, as far as I could see authors used only one metagenomic assembly software, and only one metagenomic binning workflow. It is unclear how variations in the bioinformatics workflow for assembly and binning impacts the genomic regions determined as missing. As a result, authors can only compare culture representatives with one single metagenomic workflow. I would appreciate an extensive answer of the authors regarding this matter. Would they consider expending their experimental design for a more comprehensive study?

Specific notes:

Introduction:

Ln 42-43: Key references of the pioneer publications supporting the sentence are missing. Please consider introducing a more relevant history of high-throughput sequencing in the context of culture-independent surveys.
Ln 58-59: Please consider reformulating the sentence, as single copy core genes are used to determine the completion, as well as the redundancy, of bins without the need for any reference genomes.
Ln 81-82: what workflow did the authors used, specifically? Many automatic, and a few manual-binning tools are available. They all provide different results.

M&M:

Please provide links to the genomic and metagenomic raw reads used for mapping. The study uses this data to assess coverage values and as a result it is important to make it easily available to the reader.

Results and discussion:

Ln 197: does “save” corresponds to “except”? Please consider using a more commonly used term for an optimal reading experience.
Ln 207: please consider using a different term. Genome reconstruction could refer to the assembly, or the binning. A more specific term would be appreciated.
Ln 221-223: This is incorrect. Duplicated regions will exhibit 2-fold coverage increase across all samples, and thus be clustered with the associated genome when binned using differential coverage. On the other hand, multi-copy plasmids for instance can create problematic situations, if the regulation of copy-numbers changes across samples. Please reformulate the sentence.
Ln 229-230: sequence coverage and genome coverage of what? I understand the metagenomic coverage variations described later on, but not this one. Is it based on reads recruitment from the pure culture? Please explain.

Figure 3: legend does not match the figure. Was the wrong figure uploaded?

Ln 296-298: Unclear. How adding controls not related to the studied environment (generally a black box) will help understand assembly accuracy?
Ln 304: Why 2000nt?
Ln 306: The “enhanced” binning results of this reference are contaminated due to a lack of proper curation step. Please consider using a better reference.

Conclusion:

Ln 325: Please fix typo.

·

Basic reporting

## Minor

* the central figures (Fig. 2, Fig. 3) need better annotation and a legend, otherwise the reader has to spend quite some to switching between the caption text and figure to understand them

l. 59-63: other comparisons to sequenced isolates and controlled simulation benchmarks with complex communities have been made to assess properties such as quality and completeness of reconstructed genomes. The authors should mention and cite some of them.

l. 72: explain axenic, this is a special term mostly unknown to non-biologist readers

l. 94-95: data set vs. dataset (should be consistent throughout the manuscript)

l. 107: "tetranucleotide frequency distance chi squared analysis": why not TNF chi square analysis/test and where is the distance? It's a little confusing because there is also a term called chi2 distance/divergence.

l. 108: typo in "a custom perl scripts"

l. 113: "absolute distance" == Euclidean distance? If needed, provide formula

l. 119: explain shortly what "per-base redundancy" means

l. 121: what is the "arithmetic distance" here? If needed, provide formula

Experimental design

## Major

l. 81-82: The genome reconstruction alias binning process is treated as a black box, but there are pronounced difference between different binning procedures (see e.g. http://biorxiv.org/content/early/2017/01/09/099127). The process described as "current standard genome reconstruction techniques" is very nebulous and hides the complexity of genome reconstruction. In general, the details of the metagenome sequencing and in-silico processing (assembly, binning) are nowhere described although they represent the actual subject of the study. The choices made here have direct implications on the validity of the findings. Therefore, the authors must disclose which technology (sequencing platform, read lengths, insert size etc.) and algorithms (assemblers, contigs lengths, manual inspections etc., binning program, binning procedure, binning features) their results will relate to.

* Please provide scripts used in the calculations for the distances and p-values together with a minimal documentation and a usage/license statement.

# Minor

* When the authors generate empirical null distributions for the p-value calculation using random draws (for instance l. 112,129, ), it is not entire clear to me what pool they sample from. Is this the same genome including MDRs and CDRs?

Validity of the findings

## Major

In general, the overall findings confirm what is generally known for genome binning and add additional facts on the functional level. However, results are presented being universal although this extrapolation cannot be made without looking at different data and algorithms.

* It remains unresolved whether the missing regions are a result of an incomplete assembly or the binning. For both steps, there exists a multitude of different algorithms which lead to different output. In l. 151 the authors write that the assembly contained no errors, but it is not clear whether this also relates to missing regions. In l. 188 they write, that that differing GC content in MDRs and CDRs are due to the assembly but unless verified, it could also be an artifact of the sequencing itself.

* In l. 212 and following the authors elaborate on the effects of differential coverage binning. Basically, if absolute counts are used for binning, collapsed regions are missed because they have higher coverage. However, many binners also group such contigs by using information which is not dependent on absolute counts, such as covariance/correlation distances. For instance, the program MetaBAT recruits shorter contigs to existing binss at the end of the binning procedure. The authors must disclose the details of genome binning, otherwise the results are hard to interpret or validate. The authors should also show that their findings are not an artifact of a specific binning program or setting.

---

## Round 0.2 · accepted · Accept

The most recent version reads well and should be ready to move forward. I have included some markup of the most recent version with suggested areas needing attention. I feel this work provides a good overview of considerations that need to be addressed in meta-genome reconstruction efforts. I will provide my accept decision to be considered by the section editor. Thank you for your contribution and congratulations on your efforts.